# Enhanced Optical and Antibacterial Activity of Hydrothermally Synthesized Cobalt-Doped Zinc Oxide Cylindrical Microcrystals

**DOI:** 10.3390/ma14123223

**Published:** 2021-06-11

**Authors:** Awais Khalid, Pervaiz Ahmad, Abdulrahman I. Alharthi, Saleh Muhammad, Mayeen Uddin Khandaker, Mohammad Rashed Iqbal Faruque, Abdulhameed Khan, Israf Ud Din, Mshari A. Alotaibi, Khalid Alzimami, Abdulrahman A. Alfuraih, David A. Bradley

**Affiliations:** 1Department of Physics, Hazara University Mansehra, Khyber Pakhtunkhwa 21300, Pakistan; saleh@hu.edu.pk; 2Department of Physics, University of Azad Jammu and Kashmir, Muzaffarabad 13100, Pakistan; 3Department of Chemistry, College of Science and Humanities, Prince Sattam Bin Abdulaziz University, P.O. Box 173, Al-Kharj 11942, Saudi Arabia; a.alharthi@psau.edu.sa (A.I.A.); i.din@psau.edu.sa (I.U.D.); alosaimi@psau.edu.sa (M.A.A.); 4Center for Applied Physics and Radiation Technologies, School of Engineering and Technology, Sunway University, Bandar Sunway 47500, Selangor, Malaysia; mayeenk@sunway.edu.my (M.U.K.); d.a.bradley@surrey.ac.uk (D.A.B.); 5Space Science Centre, Universiti Kebangsaan Malaysia (UKM), Bangi 43600, Selangor, Malaysia; rashed@ukm.edu.my; 6Department of Biotechnology, University of Azad Jammu and Kashmir, Muzaffarabad 13100, Pakistan; abdulhameed.khattak81@gmail.com; 7Department of Radiological Sciences, College of Applied Medical Sciences, King Saud University, P.O. Box 10219, Riyadh 11433, Saudi Arabia; kalzimami@ksu.edu.sa (K.A.); aalfuraih@ksu.edu.sa (A.A.A.); 8Department of Physics, University of Surrey, Guilford GU2 7XH, UK

**Keywords:** cobalt-doping, microcrystals, absorption, antibacterial, synthesis

## Abstract

Cobalt (Co) doped zinc oxide (ZnO) microcrystals (MCs) are prepared by using the hydrothermal method from the precursor’s mixture of zinc chloride (ZnCl_2_), cobalt-II chloride hexahydrate (CoCl_2_·6H_2_O), and potassium hydroxide (KOH). The smooth round cylindrical morphologies of the synthesized microcrystals of Co-doped ZnO show an increase in absorption with the cobalt doping. The antibacterial activity of the as-obtained Co-doped ZnO-MCs was tested against the bacterial strains of gram-negative (*Escherichia coli*, *Klebsiella pneumonia*) and gram-positive bacteria (*Staphylococcus aureus*, *Streptococcus pyogenes*) via the agar well diffusion method. The zones of inhibition (ZOI) for Co-doped ZnO-MCs against *E. coli* and *K. pneumoniae* were found to be 17 and 19 mm, and 15 and 16 mm against *S. Aureus* and *S. pyogenes*, respectively. The prepared Co-doped ZnO-MCs were thus established as a probable antibacterial agent against gram-negative bacterial strains.

## 1. Introduction

The difference of dimensions seen between the atomic and molecular scale of basic science and the microstructural scale of engineering and fabrication is balanced by the characteristic function of nanoscale materials [1]. Scientific research on crystalline nanomaterials has evolved exponentially at various levels over the last few decades. Several visualized application possibilities for such novel materials encourage intensive investigations. Semiconductors of the group (II–VI) are commonly studied because of their novel size-dependent electrical, optical, and optoelectronic properties.

Zinc oxide (ZnO) is among the most promising semiconductor materials from many perspectives [2,3,4]. It is an eco-friendly, thermally stable, biocompatible, versatile material with the potential to experience photocatalysis in a neutral, basic, and acidic medium. Such attractive properties, easy method of synthesis, and sophisticated growth develop ZnO-based devices in photonics, electronics, sensing, and acoustics [5,6]. This material is promising for optoelectronic devices of short wavelength [7,8], due to very high exciton binding energy (60 MeV), high photosensitivity, high electron mobility, inexpensive route of synthesis with diverse morphologies, flexibility in chemical functionality, biocompatibility, high transparency, and wide direct band gap (E_g_ = 3.37 eV) at room temperature [8]. Due to the huge electronegative value difference between O^2−^ (3.44) and Zn^2+^ (1.65), the bonding in ZnO is ionic. Even so, the alternating layers populated by Zn^2+^ and O^2−^ atoms form a crystal structure in which Zn^2+^ (cation) is coordinated tetrahedrally with four O^2−^ (anions). The non-centrosymmetric structure that results from this coordination (tetrahedral) produces pyroelectric and piezoelectric properties of ZnO [9]. The effective way of getting unique properties in ZnO is the introduction of impurities into the ZnO host. In the past decades, several research groups have done interesting research to work widely on the unique arrangement of transition metal (TM) ion-doped ZnO nanoparticles with magnetic and optical properties [10,11]. The TM oxide-doped ZnO holds place in various applications including sensors [12], solar cell [13], optoelectronics [14], spintronics [15], and piezoelectric devices [16]. The analysis of the variations in the properties arising due to doping of different transition metals such as Cu, Ni, Co, Mn, Cr, and Fe to ZnO has always been the matter of controversial studies [17,18,19,20,21,22,23,24,25].

Cobalt has its significance among the various TM ions due to its comparable ionic radius (0.74 Å) to that of ZnO (0.745 Å) [26]. It can change the morphology and properties of ZnO nanostructures. It has a strong magnetic moment compared to other transition metals [2]. Co-doped ZnO diluted magnetic semiconductor (DMS) may possess the ferromagnetic property at room temperature [27]. Similarly, Co-doped ZnO nanostructures are found to have more thermal stability in comparison to pure ZnO [28,29]. Co-doped ZnO structures have many potential applications in medical [30], electronics [31], photo-catalysis [32], solar cells [33], thermoelectric [34], 3D printing [35], light-emitting diodes [36], humidity sensors [37], and biosensors [38]. It can also be used as an antifungal and antibacterial agent [39]. Co-doping also makes ZnO more flexible for the above-mentioned applications compared with pure ZnO [40]. However, it is a well-known fact that structural, electrical, optical, luminescence, and magnetic properties of Co-doped ZnO are strongly dependent on the synthesis, doping, and processing techniques. Co is considered as one of the most effective elements to be doped in ZnO. In this regard, Lu et al. [41] hydrothermally prepare Co-doped ZnO nanorods to study their photocatalytic degradation. Kalpana et al. [42] prepare Co-doped ZnO by co-precipitation method and also study their photocatalytic degradation. Some other synthesis techniques have also been reported in the past, including polymeric sol-gel, polymeric precursor method, co-precipitation, auto combustion method, RF magnetron sputtering, mechanical synthesis, solvothermal synthesis, acrylamide polymerization synthesis, and thermal decomposition, to synthesize ZnO and Co-doped ZnO [43,44,45,46,47]. The impact on the band gap, in particular, its association with near band edge (NBE) emissions in Zn_1−x_Co_x_O is still unclear, as many literature studies have provided conflicting results [44]. The environmental exposure of Co is extremely high and location dependent, making it generally difficult to measure. Dietary consumption is thought to be the most common route of exposure for the general public. Cobalt is present in almost all nutrition, with the exception of vitamin B12 and other supplements. Background Co (blood) level is calculated using a normal dietary Co exposure; based on this, it is thought that it will not pose a threat to human health. Many questions about the dose-response characteristics of Co-related adverse health effects have been addressed by a newly developed biokinetic model, which shows that less than 300 µg/L blood Co concentrations are unlikely to cause clinically significant indications in healthy people. Furthermore, regular exposure at acceptable doses is unlikely to cause serious health problems [48,49,50].

Likewise, many researchers have previously investigated the bactericidal properties of ZnO nanostructures in *B. subtilis*, *S. dysenteriae*, *V. cholerae*, *E. coli*, *S. aureus*, and *S. Typhi* [50]. It has been found that the chemical interactions between membrane proteins and nanomaterials, as well as the creation of free radicals due to ZnO-NPs, might be the reason for extraordinary bactericidal properties of undoped and Co-doped ZnO-NPs. In comparison to previous work on Co-doped ZnO nanostructures [51], including nanoparticles [45,52] and nanorods [53], the current work is unique not only for the type of precursors and experimental procedures, but also for the size and cylindrical morphology of the as-obtained Co-doped ZnO, as shown in Table 1, and their effective use as a strong antibacterial agent against two (gram-positive and gram-negative) bacterial strains.

## 2. Materials and Methods

### 2.1. Materials

Zinc chloride (ZnCl_2_), cobalt-II chloride hexahydrate (CoCl_2_·6H_2_O), and potassium hydroxide (KOH) were procured from Sigma Aldrich, St. Louis, MO, USA. Reagents (99% analytically graded) were used as received without further refinement. The surface morphology of Co-doped ZnO-MCs was examined with a (QUANTA 250 FEI, FEI Company, Hillsboro, OR, USA) FE-SEM. Structural analysis of Co-doped ZnO were performed using X-ray diffractometer (Ultima IV R.I.C Tokyo, Japan), recorded at 20–80° at CuK radiation (λ = 1.54056 Å). To study the elemental composition, X-ray photoelectron spectroscopy (Thermo specific model K-α) was used. Ultraviolet–visible (UV–Vis, model T-60 Oasis Scientific Inc, Taylors, SC, USA) spectroscopy was used to study optical properties. Four separate bacterial strains were tested for antibacterial activity, namely, gram-negative; *Escherichia coli* (*ATCC^®^ 33876*), *Klebsiella pneumonia* (*ATCC^®^ BAA-1144*) and gram-positive; *Staphylococcus aureus* (*ATCC^®^ 11632*) and *Streptococcus pyogenes* (*ATCC^®^ 19615*). Nutrient agar (Oxoid^®^ CM0003) was procured from Sigma-Aldrich (St. Louis, MO, USA).

### 2.2. Synthesis of Co-Doped ZnO Microcrystals

Highly crystalline ZnO cylindrical microcrystals with Co doping were synthesized hydrothermally by using zinc chloride (ZnCl_2_), cobalt-II chloride hexahydrate (CoCl_2_·6H_2_O), and potassium hydroxide (KOH) as precursors. At first, 1.6 g of ZnCl_2_ and 0.4 g of CoCl_2_·6H_2_O were mixed. Cobalt II chloride hexahydrate is equal to the 25% of the total weight of zinc chloride. This mixture was then dissolved in 40 mL of distilled water (DI). Subsequently, 0.8 g KOH aqueous solution was added dropwise to get a precipitate with a pale pink color. Consequently, at room temperature, the solution was stirred for 30 min then the homogenous mixture was poured in an autoclave (Teflon lined) and kept in an oven for 23 h at 120 °C. The as-obtained precipitate was first washed numerous times with distilled water (DI) and then with ethanol and finally dried at 90 °C, for 30 min. The schematic representation of the whole process is demonstrated in Figure 1.

### 2.3. Screening of Antibacterial Activity

To assess the efficiency (antibacterial) of Co-doped ZnO-MCs in Mueller–Hinton broth media (from their pure cultures), all bacterial strains were sub-cultured and incubated overnight. Fresh cultures were used in the antibacterial assay by moving stock suspensions mounted on nutrient agar and incubated at 37 °C for a complete day. The bacterial culture turbidity was measured by using the 0.5 McFarland standard [58], which equals 1.5 × 10^8^ (CFU/mL) bacteria. A sterile glass spreader was used to spread each species on a Mueller–Hinton Petri dish. Wells (4 mm) were made by using a sterile polystyrene tip. Co-doped ZnO-MCs were prepared in 2% hydrochloric acid (HCL) with different concentrations (0.1, 0.5, 1 mg/mL). Forty microliters (40 µL) taken from the prepared stock solution was added to each well. Plates were placed entirely for incubation at 37 °C in an incubator overnight. In a UV transilluminator, photoactivation of Co-doped ZnO-MCs suspensions was carried out by exposure at 254 nm with UV light, for 30 min. The inhibition zone was measured around each well in millimeters by a caliper. Clindamycin phosphate, as a reference antibiotic (standard), was used at a concentration of 20 µg/mL. Every experiment was performed three times and mean value was calculated.

## 3. Results and Discussion

Figure 2a–c shows the field emission scanning electron microscopy (FE-SEM) results for the size and morphology of the as-synthesized Co-doped ZnO-MCs. Figure 2a shows the lower magnification FE-SEM micrograph. Here, individual or isolated microcrystals can be seen or observed randomly aligned in different directions. Along with randomly aligned microcrystals, some smaller species or undeveloped structures are also visible. Figure 2b shows the high magnification FE-SEM micrograph to observe the smaller as well as the larger size microcrystals. The larger size crystals are found to have a length of up to 30 μm, whereas the smaller size crystal can be found in the range of a few to greater than 5 μm. The smaller size crystals are mostly the undeveloped species observed in the previous micrograph in lower magnification. However, some undeveloped species can still be seen. Figure 2c shows the higher magnification micrograph. It confirmed the similar cylindrical morphology of all the crystals in the sample. The larger size cylinder has a diameter of 2 µm, whereas the smaller size is found to have a diameter in the range of 500 nm–1 µm. Smaller size particle-like structures can be seen stuck with the fine and smooth surface of the crystals. It points towards the stages-wise growth and Co-doping of the as-synthesized crystals. The growth and doping start from smaller particle-like structures and developed into smaller cylindrical structures with their condensation due to the increased growth duration. This condensation of the Co-doped ZnO continues with a longer growth duration of 23 h. It means that the size of the crystals can easily be adjusted by optimization of the growth duration.

XRD pattern of Co-doped ZnO-MCs is observed and shown in Figure 3. It is an analytical technique used to determine the size and nature of the material. Intensities and position of the peaks were used to assess the crystallite size, structure, and phase of the material. The miller indices have been calculated for each diffraction peak that confirms the formation of Co-doped ZnO-MCs in the sample. Miller indices for each diffraction peak is (100), (002), (101), (102), (110), (103), (200), (112), (201), (004), and (202), respectively. Crystalline phases were identified using X’Pert HighScore (version: 2.0a (2.0.1), year: 2004, manufacturer: PANalytical B.V., Almelo, Netherlands) and compare the diffraction pattern of the sample with the reference database from inorganic crystal structure database (ICSD) card/ file no: 01-076-0704. Some low intensity peaks were also observed in the sample, which is attributed to the existence of Co-based oxides (Co_2_O_3_ and CoO). The antiferromagnetic nature of Co_2_O_3_ results in a decrease in Co atomic percentage and sample’s magnetization. Cubic rock salt and hexagonal wurtzite are the two stable phases of cobalt (II) oxide CoO. Given that Zn^2+^ and Co^2+^ have ionic radii of 0.74 and 0.745, and that CoO can also crystallize in the hexagonal structure, doping and the change in NP size should have no significant impact on the lattice parameters. A change in parameters is observed due to an increase in Co concentration in ZnO [59].

The synthesized cylindrical microcrystals of Co-doped ZnO were characterized via XPS to confirm its elemental contents and chemical bonding states. The XPS results are displayed in Figure 4a–d. The full range XPS survey for the synthesized microcrystals shows several peaks (tagged for their respective elements: Zn, C, O, and Co) at different values of binding energies. The C 1s peak at 285 eV corresponds to the unavoidable carbon contamination, which might have occurred due to exposure of the sample in the air before its XPS characterization. The high-resolution XPS spectra for Zn 2p peaks are shown in Figure 4b. The first peak observed at 1021.5 eV corresponds to Zn 2p_3/2_, whereas the second peak observed at 1044.5 eV corresponds to Zn 2p_1/2_, respectively. Both the peaks in the Zn 2p states have a difference of 23 eV, which, according to the available literature, confirms the Zn^2+^ chemical states [53,60]. Figure 4c shows the high-resolution O 1 s spectrum with a single peak centered at 530.2 eV. It corresponds to lattice oxygen surrounded by zinc and cobalt ion in the hexagonal wurtzite structure [60]. Similarly, the Co 2p XPS wide scan is shown in Figure 4d with Co 2p_3/2_ peak centered at 780.5 eV and Co 2p1/2 peak at 796.5 eV. Both Co 2p peaks have a binding energy difference of 16 eV, which corresponds to the existence of Co^2+^. Thus, the XPS analysis demonstrates the doping of Co^2+^ in ZnO lattice by the substitution of Zn^2+^ with no further impurities [53].

Defects, dopant incorporation, and lattice disorder in the as-synthesized Co-doped ZnO-MCs lattice have been analyzed via non-destructive Raman spectroscopy [61]. The as-obtained Raman spectrum shown in Figure 5 has two main peaks at 97.5 and 433.5 cm^−1^, respectively. The above-mentioned peaks correspond to nonpolar E_2_ (low) and E_2_ (high) optical phonon modes of Co-doped ZnO. Unlike pure ZnO, these peaks are shifted towards higher frequencies [62]. The high-intensity peak at 382.5 cm^−1^ is assigned to the A_1_ (TO) mode of vibration [63]. The peaks observed at 162 cm^−1^ can be assigned to defect-induced mode [64], whereas the one seen at 282 cm^−1^ corresponds to B_1_ (low) phonons [65,66]. The Raman spectrum of the Co-doped ZnO also has a peak at 538.5 cm^−1^, which according to the available literature is persuaded by the host lattice defects such as Zn interstitials and oxygen vacancies. This, in other words, easily justifies the Co^2+^ doping into ZnO lattice [63,67].

Figure 6 shows UV-vis absorption spectra of Co-doped ZnO-MCs. The absorption properties are recorded in the 200–800 nm range. The value of absorption is solely determined by various types of factors such as the size and defects in grain structure. At around 374 nm, there is a rapidly rising absorption edge due to exciton recombination or defects. At higher levels of cobalt doping, the rate of absorption increases. The increase in absorption in the visible region depends on the increase in the concentration of defects causing deep levels in the ZnO band gap [68]. The increase in absorption of light is observed due to increase in lattice defects by cobalt concentration and replacing Zn^2+^ ions with Co^2+^ ions in the ZnO lattices [69].

Figure 7 shows PL spectra of Co-doped ZnO-MCs. The PL spectra show a UV near-band edge emission and blue-green emission peaks of around 373 nm and 483 nm wavelengths, respectively. The emission peak (~380 nm) is due to band-to-band excitons’ transition, while the peak at 485 nm is caused by the transition of electrons from the level of the ionized oxygen vacancies to the valence band [70]. Peak observed at ~373 nm is due to the emission from the band edge by radiative annihilation of excitons. These are linked to the recombination of excitons that are both free and shallowly bound [71,72]. The peak observed appearing at 483 nm is attributed to the formation of hydroxyl radicals and surface defects.

The antibacterial efficacy of synthesized Co-doped ZnO-MCs is assessed by agar method [73] against both gram-negative and gram-positive strains. The ability of ZnO materials to kill bacteria is normally determined by reactive oxygen species [74]. The hydrogen peroxide molecules, hydroxyl radical, and superoxide belong to the group of reactive oxygen species (ROS), which not only can cause DNA damage, but also can cause cell death [75,76]. Photocatalytic formation of ROS was the main factor in the antibacterial activity of various metal oxides [77]. Raghupati et al. [74] showed that an increase in the antibacterial activity of ZnO was associated with an increase in the production of ROS from ZnO under the influence of UV radiation. It is known that ZnO has a band gap energy of about 3.2 eV and consequently its excitation is limited to the UV radiation range. Electron-hole pairs are formed when ZnO nanostructures are exposed to UV or visible light. From Co-doped ZnO-MC suspension, these photo-generated holes separated water molecules into H+ and OH- ions. Oxygen species (dissolved) are condensed to superoxide anions (•O-2), which react with H+ to generate (HO2•). The hydrogen peroxide (H_2_O_2_) molecules are formed as they collide with the hydrogen ions and majority charge carriers (e-). The hydrogen peroxide molecules produced can penetrate the cytoplasmic membrane, killing the bacteria [78,79,80]. The schematic illustration for the antibacterial mechanism of Co-doped ZnO-MCs is demonstrated in Figure 8. ROS in large amount is produced during various photocatalytic processes. As a result, subcellular damage, including DNA damage, membrane damage, and protein denaturation, appear. Co-doped ZnO-MCs have greater antibacterial activity as a result of improved binding forces and the formation of free radicals in the cell [81,82].

All bacterial strains in the current study were recognized by using different biochemical tests conferring to the described method [83]. Pure bacteria culture was stored in a freeze-dried atmosphere at 4 °C in agar slants until further use. Figure 9 and Figure 10 show the antibacterial effects of the pure and Co-doped ZnO-MCs, against four strains, of which two are gram-negative (*E. coli*, *K. pneumonia*) and two are gram-positive (*S. aureus, S. pyogenes*). The bacterial isolates were treated with varying doses of ZnO and Co-doped ZnO-MCs (0.1, 0.5, and 1 mg/mL) dissolved in HCl (3%). Our outcomes show that, for all the tested doses, the growth of all the microbes is inhibited by both ZnO and Co-doped ZnO-MCs. An increase in ZOI with the increase in the Co-doped ZnO-MCs concentration is observed in Figure 11a,b. According to our results, gram-negative microbes, in comparison with gram-positive microbes, are more sensitive to Co-doped ZnO-MCs. Among gram-negative microbes, *E. coli* forms 17 ± 0.34 mm and 13 ± 0.26 mm ZOI, whereas *K.* pneumonia (which is more sensitive to Co-doped ZnO-MCs treatment) forms 19 ± 0.38 mm and 14 ± 0.28 mm ZOI, as shown in Table 2. Among gram-positive microbes, *S. pyogenes*, forms a ZOI of 16 ± 0.32 mm and 9 ± 0.18 mm, whereas *S. aureus* forms a ZOI of 15 ± 0.30 mm and 13 ± 0.26 mm on the same dose. Bacterial resistance is found to improve significantly with the increase in the Co content. These results suggest that Co-doped ZnO-MCs can be used as an antibacterial agent in a variety of pharmaceutical and biological applications in the future.

## 4. Conclusions

A simple and cost-effective hydrothermal technique is used to synthesize the unique cylindrical Co-doped ZnO-MCs. XRD, XPS, and Raman characterizations effectively confirmed the incorporation of Co^+2^ ions into the Zn^+2^ ions lattice site. UV-vis results showed that, at a higher dopant concentration (0.4 g/0.025 M), the rate of absorption increases. The increase in absorption in the visible region depends on the increase in the concentration of defects causing deep levels in the ZnO band gap, and this broad absorption is due to d–d transition of Co^2+^ ions. Co-doped ZnO-MCs can efficiently work against gram-negative and gram-positive bacteria. The results showed that Co-doped ZnO-MCs have better antibacterial activity against gram-negative bacteria, compared with gram-positive bacteria. All the tested bacteria were inhibited by Co-doped ZnO-MCs and the inhibitory effect was dose-dependently increased. Gram-negative microbes were shown to be more sensitive to Co-doped ZnO-MCs, as compared with gram-positive microbes.

## Figures and Tables

**Figure 1 materials-14-03223-f001:**
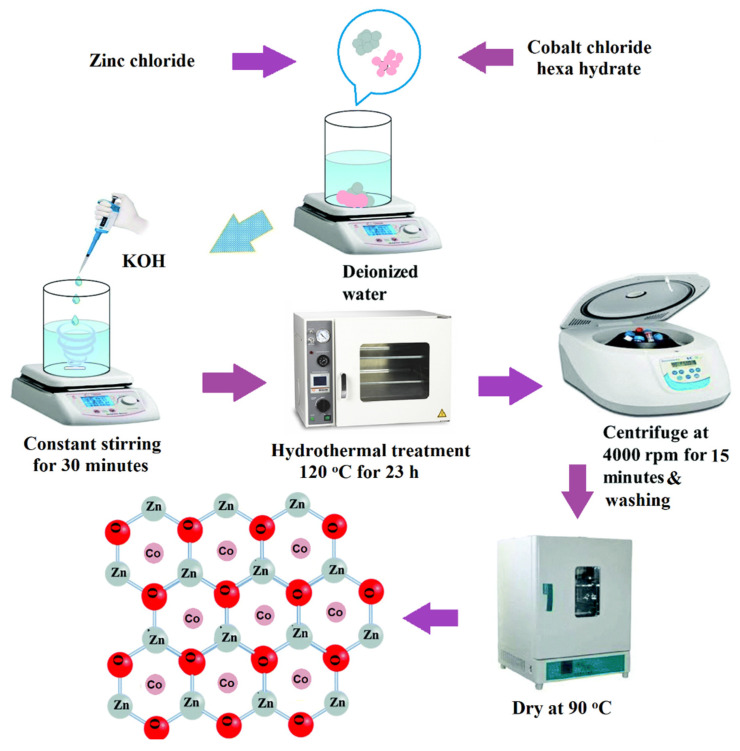
Schematic representation for the experimental setup.

**Figure 2 materials-14-03223-f002:**
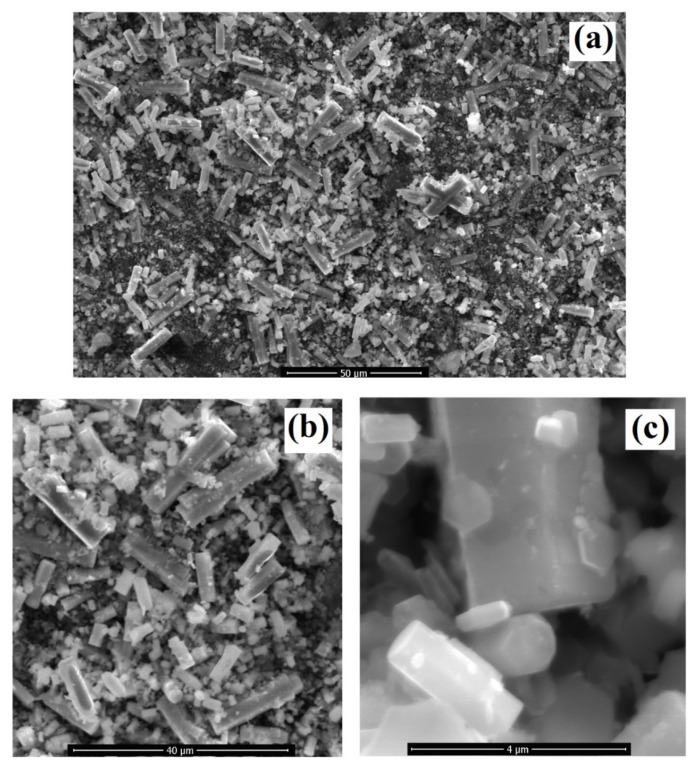
(**a**) Lower, (**b**) high, and (**c**) higher magnification micrographs of the as-synthesized Co-doped ZnO cylindrical microcrystals.

**Figure 3 materials-14-03223-f003:**
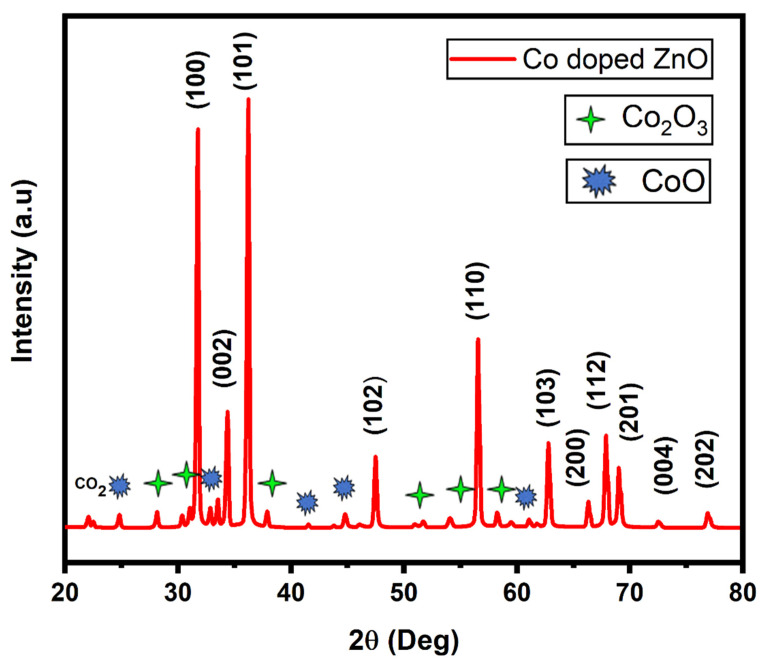
XRD pattern of the as-synthesized Co-doped ZnO microcrystals.

**Figure 4 materials-14-03223-f004:**
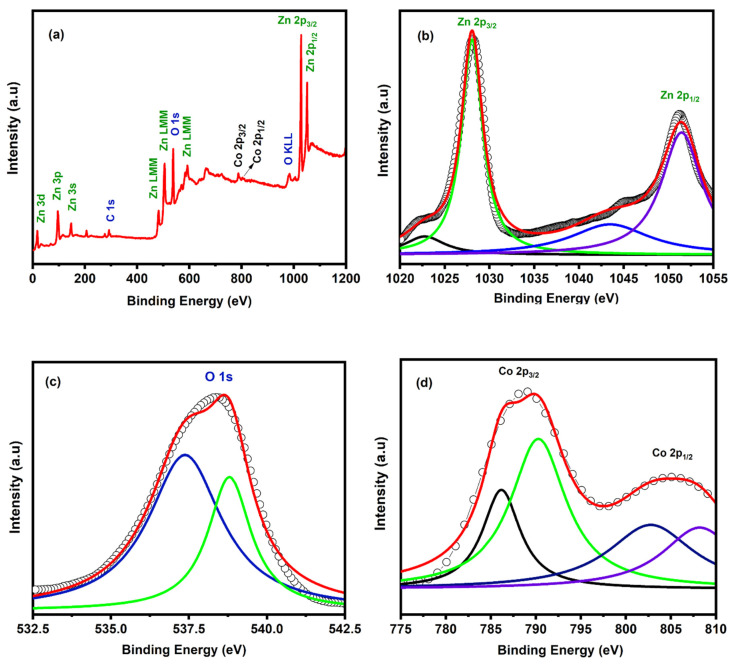
(**a**) XPS survey of Co-doped ZnO microcrystals. (**b**) High resolution XPS spectra of Zn 2p. (**c**) High resolution XPS spectrum of O 1s. (**d**) High resolution XPS spectra of Co 2p.

**Figure 5 materials-14-03223-f005:**
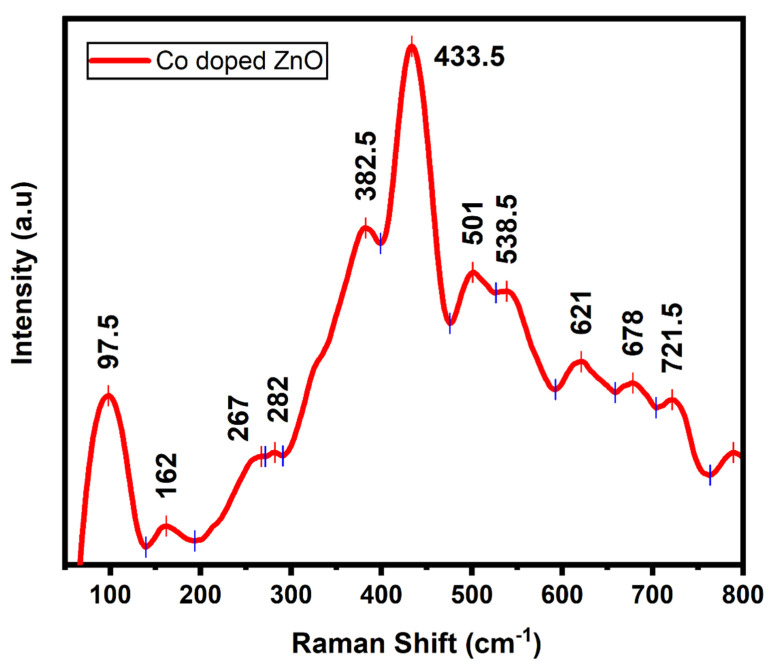
Raman spectrum of Co-doped ZnO microcrystals (MCs).

**Figure 6 materials-14-03223-f006:**
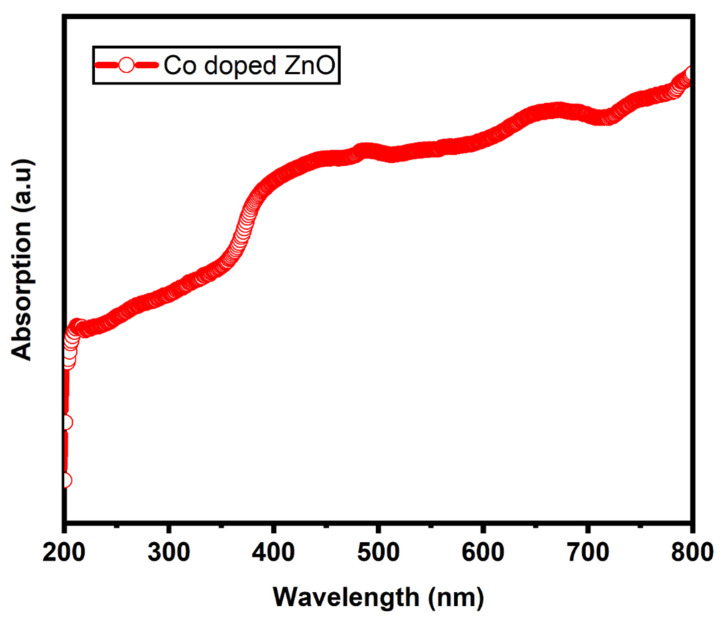
UV-vis absorption spectra of cobalt-doped ZnO microcrystals (MCs).

**Figure 7 materials-14-03223-f007:**
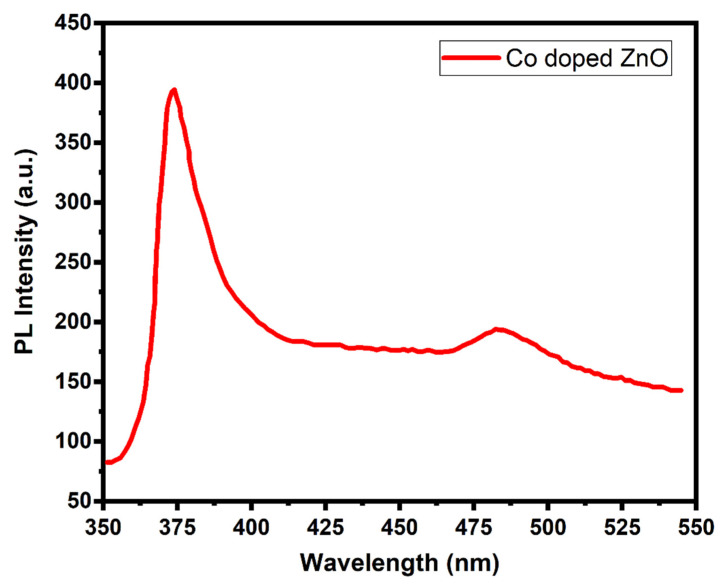
Photoluminescence spectra of Co-doped ZnO-MCs.

**Figure 8 materials-14-03223-f008:**
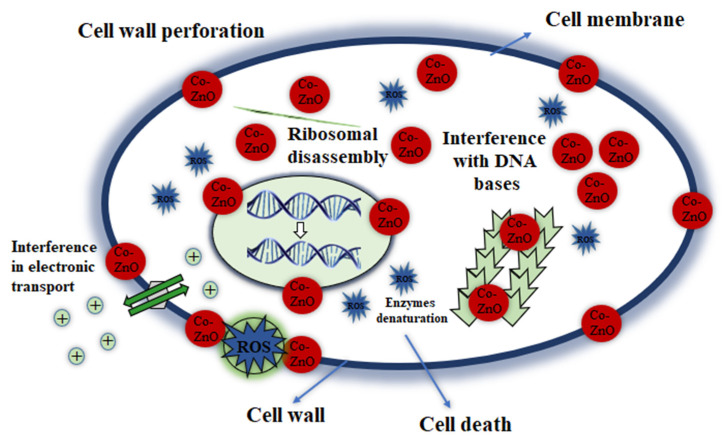
Schematic illustration for antimicrobial mechanism of Co-doped ZnO microcrystals against microbial strains.

**Figure 9 materials-14-03223-f009:**
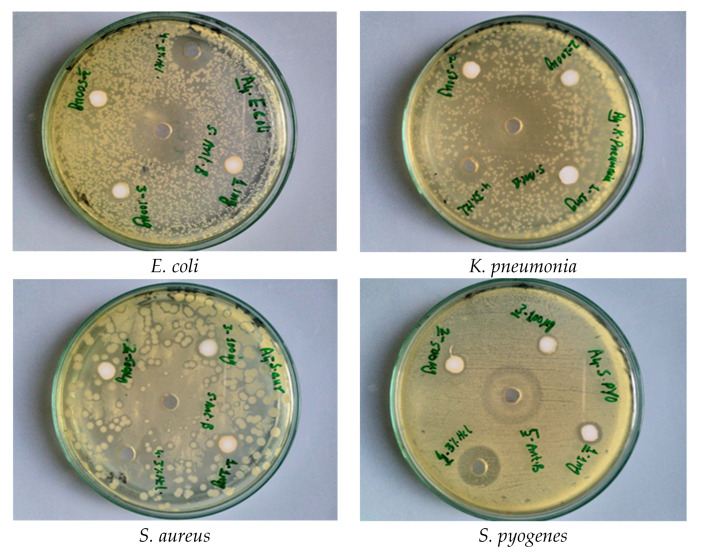
Zone of inhibition (ZOI) formed by ZnO against different bacteria.

**Figure 10 materials-14-03223-f010:**
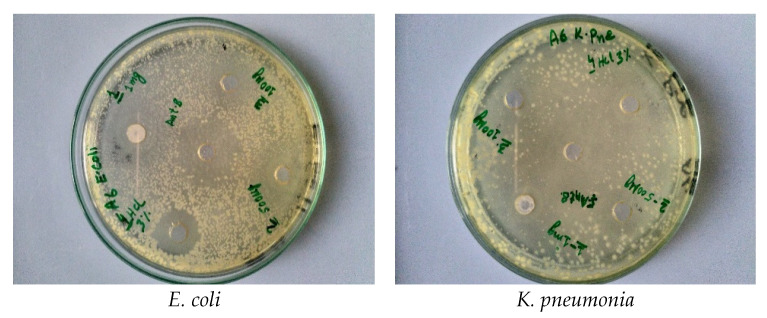
Zone of inhibition (ZOI) formed by Co-doped ZnO microcrystals against different bacteria.

**Figure 11 materials-14-03223-f011:**
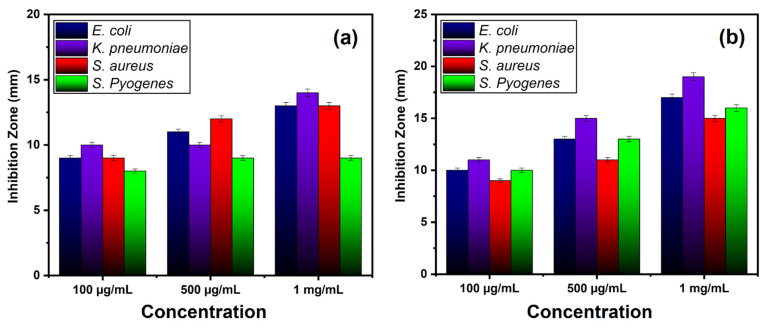
Bar graph displaying the diameter of the ZOI (in mm) produced by (**a**) ZnO and (**b**) Co-doped ZnO microcrystals (MCs) against gram-negative and gram-positive bacteria.

**Table 1 materials-14-03223-t001:** Comparison of the experimental procedure and other parameters of Co-doped ZnO-MCs with the literature.

S. No/Reference	Precursors	Temperature/Time	Technique	Morphology	Product	Confirmation	Year
[52]	Zn(NO_3_)_2_ 6H_2_O, Co(NO_3_)_2_ 6H_2_O	100 °C/16 h	Sol-gel combustion	Granular surface	Co-doped-ZnO	FE-SEM *	2014
[14]	ZnO, CoO	250 rpm/12 h	Ball milling	Nano-particles	Co-doped-ZnO	SEM **	2016
[54]	Zn(CH_3_COO)_2_·2H_2_O, Co(CH_3_COO)_2_·4H_2_O	Room temperature /3 h	Wet precipitation	Nano-particles	Co-doped-ZnO	SEM	2017
[55]	Zn(CH_3_COO)_2_·2H_2_O, Co(CH_3_COO)_2_·4H_2_O	325 K/2 h	Co-Precipitation	Nano-particles	Co-doped-ZnO	SEM	2017
[56]	Zn(CH_3_COO)_2_·H_2_O, Co(NO_3_)_2_·6H_2_O	60 °C/0.5 h	Sol-gel dip-coating	Clustered grains	Co-doped-ZnO	SEM	2017
[36]	Zn(OAc)_2_·2H_2_O, Co(II)(Acac)_2_	250 °C/0.25 h	Microwave-assisted polyol	Nano colloids	Co-doped-ZnO	SEM	2018
[57]	Zn(NO_3_)_2_·6H_2_O, Co(NO_3_)_3_·6H_2_O	95 °C/6 h	Chemical bath deposition	Nano rods	Co-doped-ZnO	SEM	2019
Our Article	ZnCl_2_, CoCl_3_·6H_2_O	120 °C/23 h	Hydro-thermal	Cylindrical microcrystals	Co-doped-ZnO	FE-SEM	2021

* Field emission scanning electron microscope, ** Scanning electron microscope.

**Table 2 materials-14-03223-t002:** Summary of the information of bacteria and other findings.

Bacteria			ZnO			Co Doped ZnO		
			100 µg/mL	500 µg/mL	1 mg/mL	100 µg/mL	500 µg/mL	1 mg/mL
Gram-negative	*E. coli*	Inhibition zone (mm)	9 ± 0.18	11 ± 0.26	13 ± 0.26	10 ± 0.2	13 ± 0.26	17 ± 0.34
	*K. pneumoniae*		10 ± 0.2	10 ± 0.21	14 ± 0.28	11 ± 0.22	15 ± 0.3	19 ± 0.38
Gram-positive	*S. aureus*		9 ± 0.19	12 ± 0.24	13 ± 0.26	9 ± 0.18	11 ± 0.22	15 ± 0.3
	*S. pyogenes*		8 ± 0.16	9 ± 0.18	9 ± 0.18	10 ± 0.2	13 ± 0.26	16 ± 0.32

## Data Availability

All the data is available within the manuscript.

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
