# Peer review of "Enhanced Optical and Antibacterial Activity of Hydrothermally Synthesized Cobalt-Doped Zinc Oxide Cylindrical Microcrystals"

_materials, 2021, doi:10.3390/ma14123223_

Round 1

Reviewer 1 Report

Dear Authors, in your interesting manuscript, the following points should be added/changed to further improve it:

  1. Abstract: There is no information about the dopant content range in ZnO.
  2. Introduction: Comment about following sentence: “The recently established biokinetic model solves many concerns related to the dose-response characteristics of Co related adverse health effects and show that blood Co concentrations of less than 300 g/l are unlikely to cause clinically significant symptoms in healthy people.” I believe that blood Co concentrations of less than 300 g/l (i.e.:gram/liters) are lethal for humans. Please provide information where 300 g/l is given as the limit value.
  3. Synthesis of Co-Doped ZnO microcrystals: There is no information given on the content of the dopant in Co-doped ZnO. Please include information about the nominal content of the dopant in ZnO (e.g. Zn(1-x)CoxO). What does the number mean (25% [135]) ?.
  4. Results and Discussion: The XRD results showed the presence of numerous foreign phases. The authors must identify the foreign phases in the sample and discuss what phases were obtained and why.
  5. Results and Discussion: Comment about following sentence: “Given that the ionic radius of Zn2+ is 0.74, whereas that of Co2+ is 0.745, and that CoO can also crystallize in the hexagonal structure, doping, i.e. the substitution of Zn2+ for Co2+ in ZnO, and the shift in NP size should not have a significant impact on the lattice parameters.” Please refer to the discussion in the literature. Doping changes the parameters of the ZnO crystal lattice (e.g. work with results concerning Co doped ZnO: DOI:10.3762/bjnano.6.200).
  6. Results and Discussion: This sentence is not understandable to me “The lattice defects increase with the cobalt concentration, and replacing Co++ ions in the ZnO lattices causes the increase in the absorption of light [52] [267-269]. Is it really about replacing Co2+ ions in the ZnO lattices?
  7. Results and Discussion: Where is the evidence for the justification of "highly crystalline nature of the as-obtained Co-doped ZnO [113]" ?
  8. Results and Discussion: Where is the evidence for the justification of “the current work is unique not only for the type of precursors [111-112]” ?
  9. Results and Discussion: Why samples (Co-doped ZnO microcrystals) in Figure 7 are not green ? What does it mean ?
  10. Conclusions: Comments to senates “UV-vis results showed that at a higher concentration of cobalt doping the rate of absorption increases” That is, how much cobalt is in ZnO ? Basic information about the composition of the sample is missing.

I appeal to the authors in the discussion to defend the uniqueness of work “…the current work is unique not only for the type of precursors, experimental procedures but also for the size, cylindrical morphology, and highly crystalline nature of the as-obtained Co-doped ZnO and their effective use as a strong antibacterial agent against two gram-negative and two gram-positive bacterial strains. [111-114]”. Please read the text several times and correct any inaccuracies.

Author Response

Reviewer 1.

Comment

Abstract: There is no information about the dopant content range in ZnO.

Author Response

The information about the dopant content is added in the abstract section of the manuscript.

Comment

Introduction: Comment about following sentence: “The recently established biokinetic model solves many concerns related to the dose-response characteristics of Co related adverse health effects and show that blood Co concentrations of less than 300 g/l are unlikely to cause clinically significant symptoms in healthy people.” I believe that blood Co concentrations of less than 300 g/l (i.e.:gram/liters) are lethal for humans. Please provide information where 300 g/l is given as the limit value.

Author Response

It's µg/l, not g/l, and this value is given in a review article, Cobalt toxicity in humans. A review of the potential sources and systemic health effects. http://dx.doi.org/doi:10.1016/j.tox.2017.05.015

Comment

Synthesis of Co-Doped ZnO microcrystals: There is no information given on the content of the dopant in Co-doped ZnO. Please include information about the nominal content of the dopant in ZnO (e.g. Zn(1-x)CoxO). What does the number mean (25% [135])?.

Author Response

Information about the dopant content is added to the manuscript. In this work, ZnO is doped with 25% cobalt.

Comment

Results and Discussion: The XRD results showed the presence of numerous foreign phases. The authors must identify the foreign phases in the sample and discuss what phases were obtained and why.

Author Response

The foreign phases present in the sample showed by XRD results are discussed in the manuscript.

Comment

Results and Discussion: Comment about following sentence: “Given that the ionic radius of Zn2+ is 0.74, whereas that of Co2+ is 0.745, and that CoO can also crystallize in the hexagonal structure, doping, i.e. the substitution of Zn2+ for Co2+ in ZnO, and the shift in NP size should not have a significant impact on the lattice parameters.” Please refer to the discussion in the literature. Doping changes the parameters of the ZnO crystal lattice (e.g. work with results concerning Co doped ZnO: DOI:10.3762/bjnano.6.200).

Author Response

According to the reference suggested, lattice parameters of the ZnO don't change with a minimum concentration of Co. If the Co concentration increased, then a change in lattice parameters is observed. We include it in the text and cite it with appropriate bibliographic references.

Comment

Results and Discussion: This sentence is not understandable to me “The lattice defects increase with the cobalt concentration and replacing Co++ ions in the ZnO lattices causes the increase in the absorption of light [52] [267-269]. Is it really about replacing Co2+ ions in the ZnO lattices?

Author Response

There is a mistake in that sentence which is corrected. The correct sentence is "increase in absorption of light is observed due to increase in lattice defects by cobalt concentration and replacing Zn2+ ions by Co2+ ions in ZnO lattice".

Comment

Results and Discussion: Where is the evidence for the justification of "highly crystalline nature of the as-obtained Co-doped ZnO [113]" ?

Author Response

The existence and sharpness of the XRD peak confirms the highly crystalline nature of the as-obtained Co-doped ZnO [Solid State Ionics: Advanced Materials for Emerging Technologies by B. V. R. Chowdari]

Comment

Results and Discussion: Where is the evidence for the justification of “the current work is unique not only for the type of precursors [111-112]” ?

Author Response

To the best of our knowledge, Co-doped ZnO cylindrical microcrystals have not been claimed by anyone in literature to date. we compare our findings with the published literature as shown in the table below.

S. No /

Reference

Precursors

Precursors Ratio

Temperature

/ Time

Technique

Morphology

Product

Confirmation

Year

[1]

Zn (NO3)2 6H2O,

Co (NO3)2 6H2O

100 °C /

16h

sol-gel combustion method

Granular surface

Co-doped-ZnO

FESEM

2014

[2]

ZnO,

CoO

~ 8:1

250 rpm / 12h

Ball milling method

Nanoparticles

Co-doped-ZnO

SEM

2016

[3]

(Zn (CH3COO)2 ·2H2O,

Co (CH3COO)2 ·4H2O

0.5 g

Room temp / 3h

Wet precipitation

Nanoparticles

Co-doped-ZnO

SEM

2017

[4]

(Zn (CH3COO)2 ·2H2O,

Co (CH3COO)2 ·4H2O

325 K / 2h

Co- Precipitation method

Nanoparticles

Co-doped-ZnO

SEM

2017

[5]

Zn (CH3COO)2. H2O,

Co (NO3)2 · 6H2O.

~ 1:1

60 °C / 0.5h

Sol-gel

dip-coating

method

Clustered grains

Co-doped-ZnO

SEM

2017

[6]

Zn (OAc)2.2H2O,

Co (II)(Acac)2

0.709 g: 0.089 g

250 °C / 0.25h

Microwave-assisted polyol

Nano colloids

Co-doped-ZnO

SEM

2018

[7]

Zn (NO3)2

·6H2O, Co (NO3)3

·6H2O

95 °C / 6h

Chemical bath deposition

Nano rods

Co-doped-ZnO

SEM

2019

Our Article

ZnCl2, CoCl3.6H2O

~ 3.2: 0.8

120 °C / 23h

Hydro-thermal

Cylindrical microcrystals

Co-doped-ZnO

FESEM

2021

Comment

Results and Discussion: Why samples (Co-doped ZnO microcrystals) in Figure 7 are not green? What does it mean?

Author Response

Co-doped ZnO microcrystals appear in a pale pink color [137].

Comment

Conclusions: Comments to senates “UV-vis results showed that at a higher concentration of cobalt doping the rate of absorption increases” That is, how much cobalt is in ZnO? Basic information about the composition of the sample is missing.

Author Response

Basic information about the sample is included in the manuscript.

Reviewer 2 Report

The manuscript reports the Antibacterial Activity of Cobalt-doped Zinc Oxide Cylindrical Microcrystals. Both the physical-chemical characterization of the synthesized MCs and the antibacterial assay have been adequately performed and presented. However, the manuscript could be accepted for publication after major revision, and the following concerns need to be addressed:

  • In the Abstract, the Authors propose the synthesized Co-doped ZnO-MCs as a potential antibiotic against Gram-negative multi-drug resistant bacterial strains. However, the term antibiotic is more appropriately attributed to a drug used to treat bacterial infections. Therefore, it would be more suitable to categorize this application under the more general category of antibacterial treatment.

  • When the Authors affirm that “Co-doped ZnO structures have many potential applications in photo-catalysis, solar cells, acoustic devices, piezoelectric devices, transistors, medicines, rubber products, textiles, laser diodes, light-emitting diodes, food packaging, and biosensors” (lines 73-75), the opportune bibliographic references should be added.

  • In the lines 273-274, the Authors claim that the bactericidal property of ZnO materials is due to reactive oxygen species (ROS). However, they do not describe the ROS production mechanisms. It would be appropriate to underline this aspect by briefly mentioning the ROS formation phenomena using suitable references (for example doi:10.3390/nano10081458 and doi.org/10.1002/chem.201902886).

  • In the lines 275-277 the following statement is reported: “The hydrogen peroxide molecules, hydroxyl radical, and superoxide belong to the group of reactive oxygen species (ROS), which not only can cause DNA damage, but also can cause the cell death [54]”. The reference 54 [Tuning of the crystallite and particle sizes of ZnO nanocrystalline materials in solvothermal synthesis and their photocatalytic activity for dye degradation, The Journal of Physical Chemistry C, 115 (2011) 13844-13850] reports the degradation of dye Rhodamine B (RhB) under visible irradiation, using the ZnO nanocrystallines, but does not explain the mechanisms of cell death and DNA damage. It is advisable to briefly explain the mechanisms of cell damage induced by ROS by citing the appropriate bibliographical references.

  • The antibacterial mechanism showed in the schematic illustration in Figure 6 should be described also in the text.

  • The sentence “Large amount of ROS are produced during the reaction.” (lines 278-279) is incomplete. What are the reactions involved in the mechanism?

  • In the experimental session, the photoactivation of samples has not be described. It would be advised to specify the illumination time, the UV-visible wavelength range and the light intensity used to irradiate the studied system.

  • Some typos are present in the text. Please, pay attention.

Author Response

Comment

In the Abstract, the Authors propose the synthesized Co-doped ZnO-MCs as a potential antibiotic against Gram-negative multi-drug resistant bacterial strains. However, the term antibiotic is more appropriately attributed to a drug used to treat bacterial infections. Therefore, it would be more suitable to categorize this application under the more general category of antibacterial treatment.

Author Response

We have replaced the term antibiotic with antibacterial agent in the manuscript as suggested.

Comment

When the Authors affirm that “Co-doped ZnO structures have many potential applications in photo-catalysis, solar cells, acoustic devices, piezoelectric devices, transistors, medicines, rubber products, textiles, laser diodes, light-emitting diodes, food packaging, and biosensors” (lines 73-75), the opportune bibliographic references should be added.

Author Response

The bibliographic references for the said text are added in the text as suggested.

Comment

In the lines 273-274, the Authors claim that the bactericidal property of ZnO materials is due to reactive oxygen species (ROS). However, they do not describe the ROS production mechanisms. It would be appropriate to underline this aspect by briefly mentioning the ROS formation phenomena using suitable references (for example doi:10.3390/nano10081458 and doi.org/10.1002/chem.201902886).

Author Response

 ROS production mechanism is explained briefly in the text by using different references.

Comment

In the lines 275-277 the following statement is reported: “The hydrogen peroxide molecules, hydroxyl radical, and superoxide belong to the group of reactive oxygen species (ROS), which not only can cause DNA damage, but also can cause the cell death [54]”. The reference 54 [Tuning of the crystallite and particle sizes of ZnO nanocrystalline materials in solvothermal synthesis and their photocatalytic activity for dye degradation, The Journal of Physical Chemistry C, 115 (2011) 13844-13850] reports the degradation of dye Rhodamine B (RhB) under visible irradiation, using the ZnO nanocrystallines, but does not explain the mechanisms of cell death and DNA damage. It is advisable to briefly explain the mechanisms of cell damage induced by ROS by citing the appropriate bibliographical references.

Author Response

In lines, 275-277 reference 54 is removed and the mechanism of cell damage induced by ROS is included in the manuscript by citing appropriate references.

Comment

The antibacterial mechanism showed in the schematic illustration in Figure 6 should be described also in the text.

Author Response

Schematic illustration for the antibacterial mechanism is explained in the text.

Comment

The sentence “Large amount of ROS are produced during the reaction.” (lines 278-279) is incomplete. What are the reactions involved in the mechanism?

Author Response

It is well established that ROS generation and catalytic activity results from photoexcitation. The photocatalytic reaction is initiated when semiconductors absorb light having a wavelength with energy equivalent to their band gaps leading to the promotion of excited electrons from the valence band to the empty conduction band, concomitantly generating electron/hole pairs. The resulting electrons/holes with sufficient reductive/oxidative power can react with surrounding oxygen-containing species, such as dissolved oxygen or H2O (OH-) and produce reactive oxygen species.

Comment

In the experimental session, the photoactivation of samples has not to be described. It would be advised to specify the illumination time, the UV-visible wavelength range and the light intensity used to irradiate the studied system.

Author Response

For the photoactivation of samples, the illumination time and UV-visible wavelength range are added in the experimental section as suggested.

Comment

Some typos are present in the text. Please, pay attention.

Author Response

 The whole manuscript is carefully revised for typos, as suggested.

Reviewer 3 Report

In the present study, cobalt (Co) doped zinc oxide (ZnO) microcrystals are synthesized using a hydrothermal method. The obtained products are characterized by FESEM, XRD, Raman, and optical analysis. Further, the antibacterial activity of the as-obtained Co-doped ZnO is tested against the bacterial strains of Gram-negative (Escherichia coli, Klebsiella pneumonia), and Gram-positive bacteria (Staphylococcus aureus, Streptococcus pyogenes) via the agar well diffusion method. The manuscript is well organized and contains interesting findings. However, I recommended a major revision of the article from its present form before it can be published in materials. The main concerns are listed below.

  1. The authors should explain the novelty of the present report in a scientific manner?
  2. The abstract and conclusion sections should be a specific and scientific approach.
  3. The authors should provide the schematic representation for the experimental procedure.
  4. What is the pH of the reaction solution? The pH of the solution normally varies from precursor to precursor. The authors must justify the selection of pH, temperature, and time.
  5. Why authors choose only one concentration of cobalt doping. The authors should perform experiments with multiple concentrations of cobalt doping and present the results as well.
  6. The authors should confirm the doping of cobalt from HRTEM and mapping.
  7. Photoluminescence spectra are needed to explore the mechanism.
  8. The application section very poor. The authors should explain in a detailed manner.
  9. What is the key factor (e.g., surface area, chemical composition, morphology) affecting the antibacterial efficiency?
  10. In the current state, there are more typographical errors and the language should be improved. Therefore, the authors are advised to recheck the whole manuscript for improving the language and structure carefully.

Author Response

Comment

The authors should explain the novelty of the present report in a scientific manner?

Author Response

To the best of our knowledge, Co-doped ZnO cylindrical microcrystals have not been claimed by anyone in literature to date. we compare our findings with the published literature as shown in the table below.

S. No /

Reference

Precursors

Precursors Ratio

Temperature

/ Time

Technique

Morphology

Product

Confirmation

Year

[1]

Zn (NO3)2 6H2O,

Co (NO3)2 6H2O

100 °C /

16h

sol-gel combustion method

Granular surface

Co-doped-ZnO

FESEM

2014

[2]

ZnO,

CoO

~ 8:1

250 rpm / 12h

Ball milling method

Nanoparticles

Co-doped-ZnO

SEM

2016

[3]

(Zn (CH3COO)2 ·2H2O,

Co (CH3COO)2 ·4H2O

0.5 g

Room temp / 3h

Wet precipitation

Nanoparticles

Co-doped-ZnO

SEM

2017

[4]

(Zn (CH3COO)2 ·2H2O,

Co (CH3COO)2 ·4H2O

325 K / 2h

Co- Precipitation method

Nanoparticles

Co-doped-ZnO

SEM

2017

[5]

Zn (CH3COO)2. H2O,

Co (NO3)2 · 6H2O.

~ 1:1

60 °C / 0.5h

Sol-gel

dip-coating

method

Clustered grains

Co-doped-ZnO

SEM

2017

[6]

Zn (OAc)2.2H2O,

Co (II)(Acac)2

0.709 g: 0.089 g

250 °C / 0.25h

Microwave-assisted polyol

Nano colloids

Co-doped-ZnO

SEM

2018

[7]

Zn (NO3)2

·6H2O, Co (NO3)3

·6H2O

95 °C / 6h

Chemical bath deposition

Nano rods

Co-doped-ZnO

SEM

2019

Our Article

ZnCl2, CoCl3.6H2O

~ 3.2: 0.8

120 °C / 23h

Hydro-thermal

Cylindrical microcrystals

Co-doped-ZnO

FESEM

2021

Comment

The abstract and conclusion sections should be a specific and scientific approach.

Author Response

The abstract and conclusion are revised as suggested.

Comment

The authors should provide the schematic representation for the experimental procedure.

Author Response

The schematic representation for the experimental procedure is provided as suggested.

Comment

What is the pH of the reaction solution? The pH of the solution normally varies from precursor to precursor. The authors must justify the selection of pH, temperature, and time.

Author Response

The pH of the reaction solution is given in the manuscript. The selection of pH, temperature, and time are justified as suggested.

Comment

Why authors choose only one concentration of cobalt doping. The authors should perform experiments with multiple concentrations of cobalt doping and present the results as well.

Author Response

We appreciate the reviewer's suggestions, however, due to the COVID-19 intense situation our labs are closed which's why we have chosen only a single concentration.

Comment

The authors should confirm the doping of cobalt from HRTEM and mapping.

Author Response

Again we appreciate the reviewer's suggestion, however, due to the serious Covid-19 situation and lockdown in our country, almost all our labs are closed. As a result, we are not able to do this in the current situation.

Comment

Photoluminescence spectra are needed to explore the mechanism.

Author Response

Photoluminescence spectra are included in the manuscript as per the reviewer's suggestion.

Comment

The application section very poor. The authors should explain in a detailed manner.

Author Response

The application section is improved and explained in a detailed manner as suggested.

Comment

What is the key factor (e.g., surface area, chemical composition, morphology) affecting the antibacterial efficiency?

Author Response

All the factors (e.g., surface area, chemical composition, morphology) affecting the antibacterial efficiency but Chemical composition and specific morphology effecting the antibacterial efficiency the most.

Comment

In the current state, there are more typographical errors and the language should be improved. Therefore, the authors are advised to recheck the whole manuscript for improving the language and structure carefully.

Author Response

The manuscript is carefully revised for grammatical mistakes by our senior Professor Dr. Mayeen Uddin Khandaker and counter-checked via Grammarly software.

Round 2

Reviewer 1 Report

  1. Comment: Abstract: There is no information about the dopant content range in ZnO.

Author Response: The information about the dopant content is added in the abstract section of the manuscript.

Review Response: First, the authors did not provide what "25%Co" mean. Whether it is a mass or quantity percentage?

Secondly, on the basis of the given masses of the reactants, I checked the given result “25%Co”. The given value is incorrect. I am asking for a thorough explanation.

  1. Comment Results and Discussion: The XRD results showed the presence of numerous foreign phases. The authors must identify the foreign phases in the sample and discuss what phases were obtained and why.

Author Response: The foreign phases present in the sample showed by XRD results are discussed in the manuscript.

Review Response: The authors identified only some of the foreign phases. Still in Figure 3, not all peaks have been identified. Please identify all crystalline phases present in the sample. Please explain why foreign phases were formed in the sample 

  1. Comment Results and Discussion: Where is the evidence for the justification of "highly crystalline nature of the as-obtained Co-doped ZnO [113]" ?

Author Response: The existence and sharpness of the XRD peak confirms the highly crystalline nature of the as-obtained Co-doped ZnO [Solid State Ionics: Advanced Materials for Emerging Technologies by B. V. R. Chowdari]

Review Response: First of all, please define what the phrase "highly crystalline nature" means. Please note that this is a scientific publication. There is no room for guesswork here, we rely on the evidence, that is, the results. Second, please tell me if the XRD analysis confirms the presence of the amorphous phase ? Third, the authors received a sample consisting of several phases. This "mixture of phases" has the properties given in the manuscript. It is worth highlighting it.

  1. Comment: Results and Discussion: Where is the evidence for the justification of “the current work is unique not only for the type of precursors [111-112]” ?

Author Response: To the best of our knowledge, Co-doped ZnO cylindrical microcrystals have not been claimed by anyone in literature to date. we compare our findings with the published literature as shown in the table below.

Review Response: Please add a table to the manuscript.

  1. Comment Results and Discussion: Why samples (Co-doped ZnO microcrystals) in Figure 7 are not green? What does it mean?

Author Response: Co-doped ZnO microcrystals appear in a pale pink color [137].

Review Response: Please add a table to the manuscript. I repeat my request. Please explain to me why this sample color is not green? What are the causes and what does it mean ?

Author Response

Comment

First, the authors did not provide what "25%Co" means. Whether it is a mass or quantity percentage?

Secondly, on the basis of the given masses of the reactants, I checked the given result “25%Co”. The given value is incorrect. I am asking for a thorough explanation.

Author Response

25% cobalt is the quantity percentage. The total amount of Cobalt II chloride hexahydrate is equal to 25% of the total weight of Zinc chloride used in this study (i.e. ZnCl2 = 1.6 g and 25 % of 1.6 g is equal to 0.4). In molar ratios 0.025 M of Cobalt II chloride hexahydrate is doped in 0.17 M of Zinc chloride (Zncl2).

Comment

The authors identified only some of the foreign phases. Still, in Figure 3, not all peaks have been identified. Please identify all crystalline phases present in the sample. Please explain why foreign phases were formed in the sample

Author Response

The remaining peaks in Figure 3 are identified and included as suggested.         

Comment

First of all, please define what the phrase "highly crystalline nature" means. Please note that this is a scientific publication. There is no room for guesswork here, we rely on the evidence, that is, the results. Second, please tell me if the XRD analysis confirms the presence of the amorphous phase? Third, the authors received a sample consisting of several phases. This "mixture of phases" has the properties given in the manuscript. It is worth highlighting it.

Author Response

We agree with the reviewer's comment that our sample consisting of several phases and we remove the phrase "highly crystalline nature" from the manuscript. The phrase was written mistakenly into the manuscript.

Comment

Please add a table to the manuscript.

Author Response

The table is added to the manuscript as suggested.

Comment

I repeat my request. Please explain to me why this sample color is not green? What are the causes and what does it mean?

Author Response

During the synthesis process, all the cobalt in the solution was not consumed that's why the sample color is not green instead shows pale pink color.

Another reason is the presence of different concentrations of the deep level defects such as oxygen/zinc vacancies and oxygen/zinc interstitials introduce different deep energy levels in the bandgap These defects follow rules of thermodynamics. And if their number is large, they form a band. Excitons too could form a band like a defect band. In the forbidden gap. The color depends on where the band is located. it might be a case of absorption of ambient or instrument light and re-emitting it.

Reviewer 2 Report

The Authors adequately addressed the criticisms arisen in the first revision. However, I consider that the following points should be considered in order to improve the manuscript quality:

  • In lines 263-265, the Authors report that “Raghupati et al. [71] showed that an increase in the antibacterial activity of ZnO was associated with an increase in the production of ROS from ZnO under the influence of UV radiation.” I would like to underline that reference 71 does not correspond to Raghupati's work and does not related to the subject under discussion. Pay attention to this point.                       Furthermore, the concept reported in this sentence should be better specified. It is known that ZnO have a band gap energy of about 3.2 eV and consequently its excitation is limited to the UV radiation range.

  • About my previous comment (The sentence “Large amount of ROS are produced during the reaction.” (lines 278-279) is incomplete. What are the reactions involved in the mechanism?), I agree with the exhaustive response of the Authors.                                                             However, I believe that the phrase “ROS in large amount is produced during the reaction” reported in the manuscript is not very exact. It would be more appropriate to speak of ROS produced by different photocatalytic processes rather than a reaction.

  • Some typos are still present in the text.

Author Response

Comment

In lines 263-265, the Authors report that “Raghupati et al. [71] showed that an increase in the antibacterial activity of ZnO was associated with an increase in the production of ROS from ZnO under the influence of UV radiation.” I would like to underline that reference 71 does not correspond to Raghupati's work and does not related to the subject under discussion. Pay attention to this point.                       Furthermore, the concept reported in this sentence should be better specified. It is known that ZnO have a band gap energy of about 3.2 eV and consequently its excitation is limited to the UV radiation range.

Author Response

The correct reference related to Raghupati et al. work is cited in the related text under discussion as suggested.

Comment

About my previous comment (The sentence “Large amount of ROS are produced during the reaction.” (lines 278-279) is incomplete. What are the reactions involved in the mechanism?), I agree with the exhaustive response of the Authors.                                                             However, I believe that the phrase “ROS in large amount is produced during the reaction” reported in the manuscript is not very exact. It would be more appropriate to speak of ROS produced by different photocatalytic processes rather than a reaction.

Author Response

The phrase “ROS in large amount is produced during the reaction” is changed as suggested.

Comment

Some typos are still present in the text.

Author Response

We thoroughly studied the manuscript and correct the typos present in the text.

Reviewer 3 Report

The manuscript can be acceptable in the present form.

Author Response

Comment

The manuscript can be acceptable in the present form.

Author Response

Thank you so much for your critical review on our article.